# Mending cracks atom-by-atom in rutile TiO$_2$ with electron beam radiolysis

Silu Guo [1], Hwanhui Yun [1,2], Sreejith Nair[1], Bharat Jalan[1] & K. Andre Mkhoyan [1] ✉

Rich electron-matter interactions fundamentally enable electron probe studies of materials such as scanning transmission electron microscopy (STEM). Inelastic interactions often result in structural modifications of the material, ultimately limiting the quality of electron probe measurements. However, atomistic mechanisms of inelastic-scattering-driven transformations are difficult to characterize. Here, we report direct visualization of radiolysis-driven restructuring of rutile TiO$_2$ under electron beam irradiation. Using annular dark field imaging and electron energy-loss spectroscopy signals, STEM probes revealed the progressive filling of atomically sharp nanometer-wide cracks with striking atomic resolution detail. STEM probes of varying beam energy and precisely controlled electron dose were found to constructively restructure rutile TiO$_2$ according to a quantified radiolytic mechanism. Based on direct experimental observation, a "two-step rolling" model of mobile octahedral building blocks enabling radiolysis-driven atomic migration is introduced. Such controlled electron beam-induced radiolytic restructuring can be used to engineer novel nanostructures atom-by-atom.

When energetic electrons interact with crystals, a combination of elastic and inelastic interactions take place. Some inelastic interactions result in displacement of the atoms from their original crystal lattice sites. These inelastic interactions often ultimately limit the precision of measurements with electron probes. Transmission electron microscopes are strongly limited by beam damage, as they commonly use high-energy (60 to 300 keV) electron beams to study atomic structure and electronic properties of crystalline materials[1]. For an atom to be displaced from its lattice site under electron beam irradiation, some (or all) of its chemical bonds must be broken. The mechanisms by which these bonds are broken are either by direct transfer of energy and momentum from incident electrons to the atoms, or by ionization of those bonding electrons, or by radiolysis[2,3]. When the energy of electrons in the incident beam is considerably high (~200–300 keV) and crystal consists of light elements (Z ≲ 20) with low binding energies (≲ 1-2 eV/bond), the knock-on mechanism dominates, drilling holes in samples[4,5]. In all other cases, the radiolytic mechanism dominates, often amorphizing crystalline samples[2,3,6] (in rare cases both

mechanisms are in play). Radiolysis requires formation of electron-hole pairs (excitons) with sufficiently long lifetime (≥1 ps) and high energy (≳2-3 eV) to provide the energy and momentum needed to break the bonds and separate atoms from each other before they can recombine[2]. It should be noted that there are distinct differences between structural changes in a material due to radiolysis and those due to sample heating, which occur under specific conditions[1].

The radiolytic bond breakage and associated crystal amorphization is well documented in many materials including halides[7,8], silicates[9,10], zeolites[11–13], and, recently, in MOFs[14,15]. The opposite was also observed, when amorphous material turns into crystal[16–18]. However, the physical processes behind atomistic mechanisms of radiolysis-driven structural transformations are still not well understood. The main obstacle is the difficulty to visualize the atomic movements in the structure when materials transform between crystalline and amorphous phases. This study sheds light on atomistic processes by directly observing a crystal-to-crystal transformation driven by radiolysis. We explored several wide band gap oxides (SiO$_2$,

[1]Chemical Engineering and Materials Science, University of Minnesota, Twin Cities, Minneapolis, MN 55455, USA. [2]Korea Research Institute of Chemical Technology, Daejeon 34114, Korea. ✉e-mail: mkhoyan@umn.edu

$Al_2O_3$, $TiO_2$, $GeO_2$, $SnO_2$, etc.) that host high-energy excitons ($\gtrsim 2$ eV) and exhibit crystal structures containing tetrahedral or octahedral building blocks, as the movements of these building blocks can be easier to track compared to those of individual atoms. While many of these oxides meet the requirements for radiolysis under electron beam irradiation, i.e., formation of long lifetime and high energy excitons, we focused on rutile $TiO_2$. This material has a tetragonal crystal structure with space group $P4_2/mnm$ (a = 4.594 Å, c = 2.959 Å)[19,20], where each Ti atom is surrounded by six O atoms forming a distorted $TiO_6$ octahedral basic building block. Two neighboring octahedra share an edge at the base with two associated oxygens, and each unit is joined with another two through corner oxygen. Since neighboring octahedra only share edges and not faces, they might be able to rotate or move into the available open space in the structure when bonds between units are broken. Earlier reports documented that rutile $TiO_2$ is indeed susceptible to structural modifications under electron beam irradiation[21,22].

In this article, we report the results of a scanning transmission electron microscopy (STEM) study of radiolysis-driven restructuring in rutile $TiO_2$. The geometry of the sample has allowed us to directly visualize and quantitatively analyze radiolytic changes occurring in the material as a function of electron beam exposure. Analysis of atomic-resolution high-angle annular dark-field (HAADF) images and electron energy-loss spectroscopy (EELS) data shows that radiolysis can be a constructive force restructuring cracks in rutile $TiO_2$ through relocation of building units. These observations point to the constructive potential of radiolysis, which can help to improve the current techniques of crystal growth and heterostructure fabrication.

## Results and Discussion

The study is conducted on single crystal rutile $TiO_2$ with nanometer-wide cracks. Samples were prepared by growing a 26 nm $IrO_2$ film on a rutile $TiO_2$ substrate (See Method section). When rutile $IrO_2$ thin film is grown on a rutile $TiO_2$ substrate, epitaxial stresses imposed by the film on the substrate ($\varepsilon = 2.2\%$ in $\langle 110 \rangle$ direction) induce atomically sharp cracks that span micrometers along the in-plane

$\langle 110 \rangle$ crystallographic directions and propagate well into the $TiO_2$ substrate along the c-axis. Such crack formation in rutile $TiO_2$ due to an epitaxial strain is well documented[23,24]. Each crack starts with a single-atom width at its tip and widens to 3 nm at the interface of the $TiO_2$ substrate and $IrO_2$ film, leading to wedge angles of 2-3° (see Supplementary Fig. 1). These cracks and their crystallographic orientations are readily identifiable in top-view scanning electron microscopy (SEM) images (Fig. 1a). Figure 1b shows the low-magnification HAADF-STEM image of one such crack cross-section (see Method section) along with a magnified atomic-resolution image of the neighboring rutile $TiO_2$ crystalline structure. Direct comparison with simulated HAADF-STEM images suggests that these samples are approximately 50 nm thick (see Supplementary Fig. 2).

To evaluate the effects of electron beam exposure, high-resolution HAADF-STEM time-lapse images of the cracks were obtained. A set of images obtained from a crack with a wedge angle of 3.6° at a dose rate of 812 e Å$^{-2}$ s$^{-1}$ is shown in Fig. 1c (see also Supplementary Movie 1). As can be seen in Fig. 1c, the crack is self-healing with some areas almost completely filled and others only partially. At the beginning of the beam exposure, at doses $D \lesssim 6.5 \times 10^4$ e Å$^{-2}$, the crack is still atomically sharp. Starting from an electron dose of approximately $5.9 \times 10^5$ e Å$^{-2}$, formation of some crystalline structures at several locations along the crack edge are evident. As cumulative electron dose increases, the crack progressively fills.

To further verify the crystalline structure and the composition of the newly formed material in the crack, high-magnification STEM images were obtained from that region. An atomic-resolution HAADF-STEM image of the filled crack after being exposed to an electron dose of $4.1 \times 10^6$ e Å$^{-2}$ is shown in Fig. 1d. The newly formed structure inside the crack is commensurate with the rutile $TiO_2$ outside the crack (Fig. 1b). This arrangement of the atomic columns is specific to rutile $TiO_2$ in $\langle 110 \rangle$ orientation. Some fluctuations in HAADF image contrast, observed in the newly formed crystal inside the crack, can be attributed to incompletion of the filling process and to some variations in local thickness of this region (see Supplementary Fig. 2).

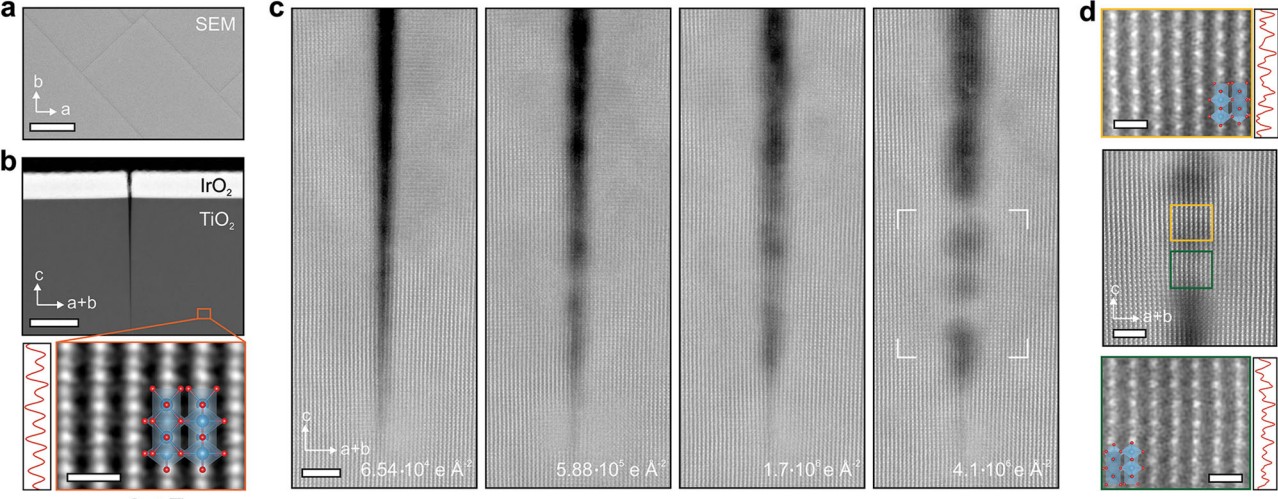

**Fig. 1 | Rutile $IrO_2/TiO_2$ sample with cracks restructuring under STEM beam.**
**a** SEM top-view image of a rutile $IrO_2/TiO_2$ sample, where the black lines running along the $\langle 110 \rangle$ crystallographic directions are the cracks. Scale bar is 1 μm. **b** HAADF-STEM image of a crack propagating through $IrO_2/TiO_2$ sample viewed along the $[1\bar{1}0]$ direction. Scale bar is 50 nm. Atomic-resolution image of the crystal structure of the rutile $TiO_2$ in this projection shows the arrangements of two distinct atomic columns: columns of only Ti atoms ("dim") and columns of combined Ti and O atoms ("bright"). The line-scan through these columns is shown on the left. Atomic model of the rutile $TiO_2$ is overlaid on the image. Scale bar is 0.5 nm.
**c** A set of HAADF-STEM images showing formation of $TiO_2$ crystal in the crack bridging two sides with increase of electron doses. Scale bar is 3 nm. **d** Atomic-resolution image of the crack region, shown in **c**, after $4.1 \times 10^6$ e Å$^{-2}$ electron dose exposure. Scale bar is 2 nm. Two magnified regions (highlighted in yellow and green) show formation of rutile $TiO_2$ crystal inside the crack viewed in $[1\bar{1}0]$ orientation. HAADF intensity profiles shown on the right are obtained from a row of atomic column. Scale bars are 0.5 nm. Images are low-pass filtered for noise reduction.

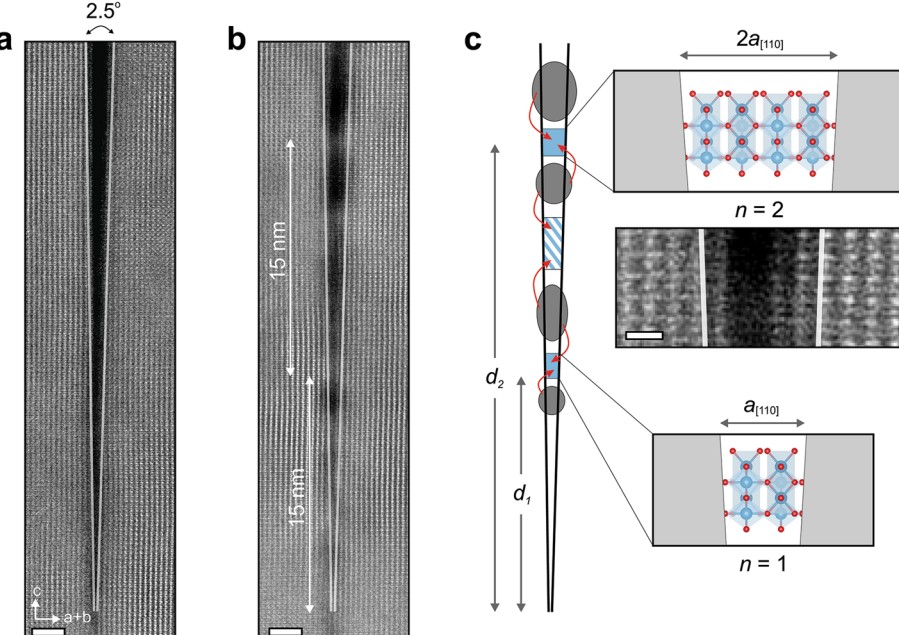

**Fig. 2 | Observation of bridging locations in the center of a crack. a** HAADF-STEM images of a crack in rutile $TiO_2$ before and **b** after exposure to the STEM electron beam. Scale bar is 2 nm. The initial bridging of the crack at specific locations are visible. The image in **b** is taken at an electron dose of $1.5 \times 10^6$ e $Å^{-2}$. **c** A model describing observed initial bridging of the crack at locations where the gap is an integer number of unit cells of rutile $TiO_2$. Scale bar is 0.5 nm. All HAADF-STEM images are low-pass filtered for noise reduction.

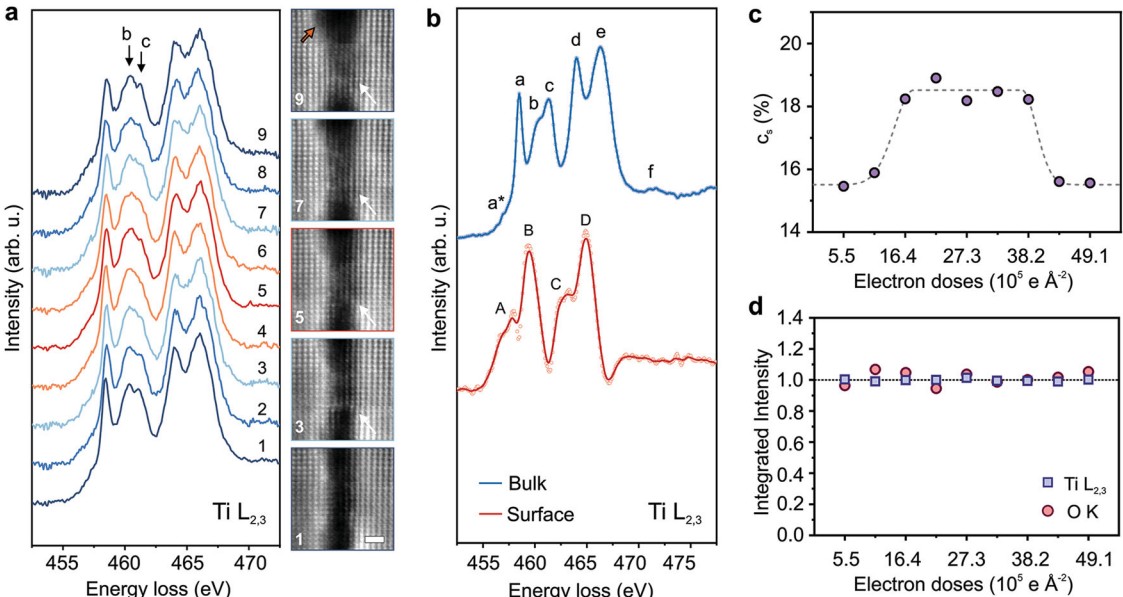

**Fig. 3 | EELS revealing the valence state changes of Ti atoms during the crack restructuring. a** A set of EELS Ti $L_{2,3}$-edges as a function of electron beam doses from a crack region in rutile $TiO_2$. HAADF-STEM images recorded in parallel with EELS measurements are presented on the right showing bridging. Scale bar is 1 nm. Images are low-pass filtered. The range of electron doses in this measurement is from $5.5 \times 10^5$ (spectrum 1) to $49.1 \times 10^5$ e $Å^{-2}$ (spectrum 9). The peaks "b" and "c" with most changes are at 460.3 and 461.1 eV, correspondingly. **b** The spectra of bulk and surface Ti $L_{2,3}$-edges with all identifiable features of their fine structure labeled as a*-f and A-D, correspondingly. **c** Concentrations of surface Ti atoms in the exposed crack area ($c_s$) as a function of electron dose determined from Ti $L_{2,3}$-edge spectra in **a** using bulk and surface references in **b**. **d** The changes in the number of Ti and O atoms as a function of electron dose in beam-exposed crack area in **a** are evaluated using integrated intensities of Ti $L_{2,3}$- and O K-edge EELS spectra. A linear fit to these data gives a slop value of $-2.93 \times 10^{-4}$ for Ti and $2.7 \times 10^{-3}$ for O, suggesting no detectable change.

A closer look at the initial restructuring stages reveals interesting details about bridging locations. At an electron dose of $1.5 \times 10^6$ e $Å^{-2}$, bridging from two opposite sides can be observed at two different locations (Fig. 2). The first bridging location is at the distance $d_1 = 15$ nm from the tip of the crack and the second one is at the same distance from the first: $d_2 = 2d_1 = 30$ nm (Fig. 2b and Supplementary Fig. 3). The gaps at these two locations are estimated to be 6.5 Å and 13.1 Å, respectively. These two gap widths are equal to an integral multiple of the unit-cell-length of rutile $TiO_2$ in this projection satisfying the relationship: $d_n \cdot \tan(\alpha) = n \cdot a_{[110]}$, where $n = 1, 2, 3, \ldots$ is

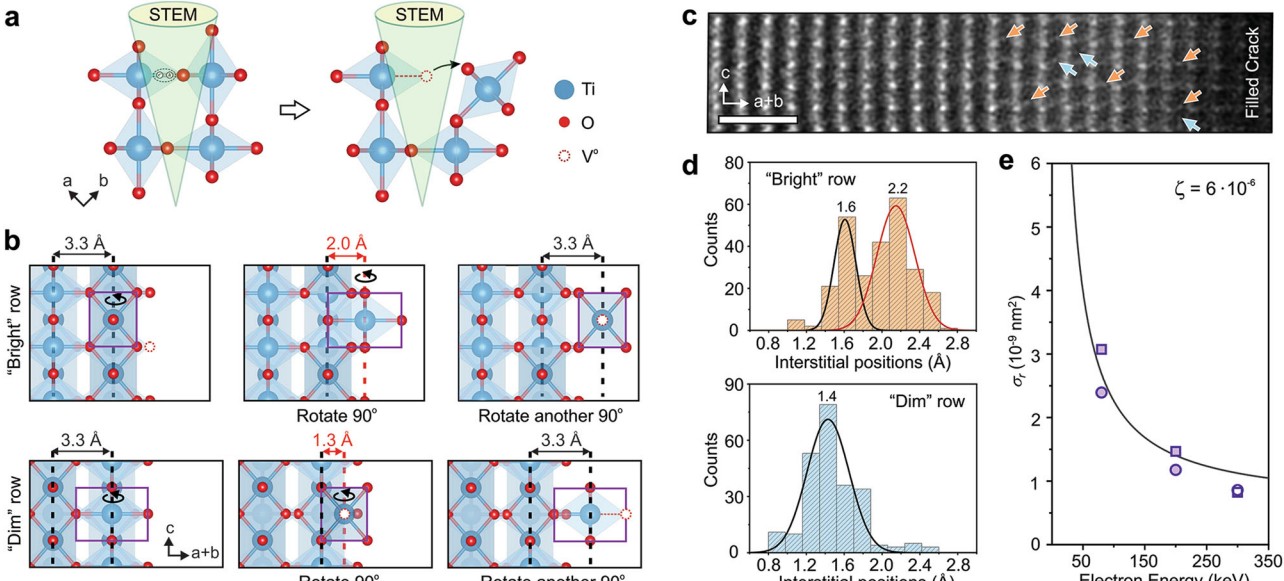

**Fig. 4 | A "two-step" rolling model for the octahedral migration driven by radiolysis. a** A schematic illustrating radiolysis-driven Ti-O bond breaking between two linking $TiO_6$ octahedra and their separation with aid of exciton ($e\text{-}h^+$ pair) generation. **b** Illustrations of the "2-step rolling" model showing how an octahedral unit (highlighted in purple) at the edge of the crack from the "bright" and "dim" atomic columns can rotate 90° and occupy an interstitial site in the first step and then reach its final site after another 90° rotation in the second step. Axes of rotation are indicated. Each interstitial site can be characterized by the distances from the nearest neighboring atomic column (highlighted by red dashed line). The

red dashed open circles in **a** and **b** indicate oxygen vacancies ($V^O$). **c** HAADF-STEM image of a partially bridged crack showing presence of Ti partial atomic columns at different interstitial sites (indicated by arrows). Scale bar is 1 nm. Image is low-pass filtered for noise reduction. **d** Statistical analysis of the locations of Ti partial atomic columns in the interstitial sites from HAADF images obtained from many "bright" and "dim" rows. **e** Radiolysis cross-section evaluation using three electron beam energies: 80, 200, and 300 keV. It is converted into probabilities of single radiolysis-driven event (one-step of rolling) and fitted. The efficiency factor is estimated to be $\zeta \simeq 6 \times 10^{-6}$.

---

positive integer number, $\alpha = 2.5°$ is crack wedge angle, and $a_{[110]} = 6.48$ Å is the unit cell width of rutile $TiO_2$ in $\langle 110 \rangle$ directions (Fig. 2c). This observation suggests that at these locations, where integer number of rutile $TiO_2$ unit cell lengths can fit, are preferable sites for initial bridging with minimal strain or defects. In between these bridging sites, some crystalline formations can also be seen, particularly at distances of $d_m \cdot \tan(\alpha) = m \cdot a_{[110]}$, where $m = \frac{1}{2}, \frac{3}{2}, \frac{5}{2}, \ldots$ (Fig. 2b). But they are not well ordered as the widths of the gaps at these locations are fractional multiples of the unit-cell-length and thus, a perfect rutile structure cannot be satisfactorily accomplished. Additionally, it is observed that the main bridging sites, at distances $d_n$, are surrounded with widened cracks from both sides (Fig. 2b), showing that material migrates from neighboring free surfaces into the bridging sites. Simplified 3D and 2D crack models are shown in Supplementary Fig. 4.

This restructuring requires extensive breaking and re-forming of Ti-O bonds near the crack. The release of Ti and O atoms enables them to move out and fill in the crack. Core-level EELS analysis was performed to understand the changes in Ti-O bonds near the crack. Core-loss EELS is particularly sensitive to changes in immediate bonding environment of the atoms in a crystal[25]. Monochromated EELS Ti $L_{2,3}$-edge fine structure was measured from the crack region as a function of electron dose. EELS spectra were recorded in parallel with HAADF-STEM images (Fig. 3a). As can be seen from these HAADF-STEM images, the crack is gradually bridged as the beam dose increases, consistent with our earlier observations. The fine structure of corresponding Ti $L_{2,3}$-edges, on the other hand, changes non-monotonically. The relative intensities of peaks "b" and "c" depart significantly from the initial fine structure before eventually reverting back. High sensitivity of these two specific peaks in Ti $L_{2,3}$-edge fine structure to changes in Ti-O bonding state is well understood. In a bulk rutile $TiO_2$ crystal, the EELS Ti $L_{2,3}$-edge exhibits two groups of peaks, $L_3$ and $L_2$, associated with electronic transitions from initial $2p_{3/2}$ and $2p_{1/2}$ core states to final $3d$ states in the conduction band[26]. In addition,

both $L_3$ and $L_2$ edges are divided into $t_{2g}$ and $e_g$ sub-bands due to octahedral crystal field splitting[27–30]. Furthermore, in the $L_3$-edge, the $e_g$ peak splits into peaks "b" and "c" due to Jahn-Teller distortion of the $TiO_6$ octahedra (with $D_{2h}$ symmetry)[28,31]. As a result, these two peaks are highly sensitive to any distortions or bond modifications in $TiO_6$ octahedra. When a Ti-O bond is broken in rutile $TiO_2$ and the O atom is left behind, the original $TiO_6$ reduces to a $TiO_5$ "octahedron" and the $Ti^{4+}$ turns into $Ti^{3+}$. This manifests in the changes in intensities and locations of peaks "b" and "c"[28,32]. By analyzing these two peaks, the changes in Ti valence-state and in Ti-O bonds are evaluated.

Since the $TiO_5$ "octahedra" with $Ti^{3+}$ are the dominant units on the surfaces of rutile $TiO_2$, first the characteristic bulk and surface Ti $L_{2,3}$-edges were obtained (Fig. 3b). They were deduced from two EELS spectra measured; one from the edge region of the crack and one from a bulk region between two cracks (for more details, see Supplementary Figs. 5 and 6). As can be seen from Fig. 3b, they are considerably different from each other due to (4 +) and (3 +) valence states of Ti in the bulk and on the surface, respectively. The fine structures of these EELS edges are consistent with those from bulk and surface Ti $L_{2,3}$-edges reported in the literature[28,32]. Using these bulk and surface EELS Ti $L_{2,3}$-edges as references, each spectrum in Fig. 3a was decomposed into bulk and surface components fitted as a linear superposition of two: $I = x \cdot I_s + (1 - x) \cdot I_b$, where $x$ and $(1 - x)$ are the relative fractions (or concentrations) of surface and bulk components (for more details, see Supplementary Fig. 7). The results are presented in Fig. 3c. The concentration of surface states initially increases to 18.5% from the initial 15.5% concentration, then recovers back to initial levels after accumulation of electron dose, indicating structural self-healing. When $TiO_x$ octahedra are moving from both sides of the crack into the gap, many Ti-O bonds are breaking, resulting in more surface-like (or $Ti^{3+}$) Ti atoms. Then, when they link to each other from opposite sides, $TiO_6$ octahedra are restored, and the region turns into a crystal with an original rutile structure. When the amounts of Ti and O atoms in this

exposed area were evaluated using integrated intensities of Ti $L_{2,3}$- and O K-edge, no detectable reduction of either Ti or O was observed (Fig. 3d), suggesting no change in composition and further supporting radiolysis-dominant filling of the gap in the cracks. It should be noted that the bulk rutile $TiO_2$ region, when exposed to these electron doses, shows no change in composition and light restructuring due to surface roughness from sample preparation (see Supplementary Fig. 8). However, at very high doses more noticeable restructuring/amorphization is observed (see Supplementary Fig. 9)

Taking all the observations discussed above into consideration, we propose the following mechanism for structural changes that take place at a crack in rutile $TiO_2$ under electron beam irradiation: sequences of "2-step rolling" bring $TiO_6$ octahedral units from both crack edges into free space to eventually bridge and fill the crack. This "2-step rolling" is illustrated in Fig. 4 (see also Supplementary Movie 2 and Supplementary Figs. 4 and 10). As illustrated in Fig. 4a, b, with radiolytic bond breakage, the $TiO_6$ octahedral units from corners, edges, and surfaces can roll and occupy new sites, thus producing net mass transfer from crack edges into the crack gap. Such rolling of the $TiO_6$ octahedra can also take place vertically, parallel to the edges of the crack (see Supplementary Fig. 10c), which accounts for the relocation of material into the gap bridging sites from nearby areas (Figs. 1, 2 and 3a). It should be noted that $TiO_6$ octahedral units often are incomplete $TiO_x$ "octahedrons" depending on number of oxygens in the unit. Formation of many new surface $TiO_x$ "octahedra" are captured in HAADF-STEM images and Ti $L_{2,3}$-edge spectra (Figs. 2 and 3). When these new octahedral units meet in the middle of the crack, they unite through corner-sharing oxygens and form the rutile $TiO_2$ crystal structure.

To further examine this octahedral "2-step rolling" model, we conducted fine-grained image analysis of electron beam irradiated areas near the cracks. Based on this rolling model, we should see many Ti atoms at interstitial sites of the rutile $TiO_2$ crystal in these regions, as the rolling octahedra pass through interstitial sites in the process of extending the crystal (as shown in Fig. 4b, also in Supplementary Figs. 4a and 10 and Supplementary Movie 2). This matches what we observed in atomic-resolution HAADF-STEM images obtained from those regions, particularly in the areas near the edges of the crack where it is bridging (Figs. 4c, d and Supplementary Fig. 11a). It should be noted that, when viewed along the ⟨110⟩ direction, there are three identifiable interstitial sites in rutile $TiO_2$ crystal where the rolling octahedra should be observed after the first rolling step: two sites between "bright" columns 1.5 and 2.0 Å away from the nearest Ti-O plane, one site between "dim" columns at 1.3 Å away from the nearest Ti-O plane. In these HAADF-STEM images of bridging regions, we identify the presence of partial Ti atomic columns in all three interstitial sites (Figs. 4c, d). The procedure used to determine the location of these interstitial units is discussed in SI (see Supplementary Figs. 11b, c).

To further evaluate the radiolysis-induced restructuring process, additional experiments with different electron beam energies were performed using similar set-up (dose rate, beam convergence angle, etc.). A set of STEM experiments were conducted at $E_O = 80$, 200, and 300 keV beam energies, using three different cracks on a single FIB-cut TEM specimen. While results show that bridging and filling occurs in all conditions, it is faster at lower beam energies (see Supplementary Fig. 12 and Movie 3), exhibiting the characteristic beam energy dependence of radiolysis. In a rutile $TiO_2$ crystal, every Ti atom is bonded to six oxygens with a bond dissociation energy of 3.3 eV/bond (see Supplementary Tables 1 and 2), where four bonds have a length of 1.94 Å and the other two (opposite to each other) are slightly longer at 1.99 Å, making them more susceptible to bond breaking[19]. Since the exciton energy in rutile $TiO_2$ is approximately 3.0 eV[33,34] with a very long exciton lifetime of $\tau_{ex} = 16$ ns[35], the formation of one or two excitons at a time readily enables radiolytic rolling of a single

octahedron. Rolling displacements of edge and corner octahedral units are expected to be more frequent than those of surface units, as they only require breakage of 1-3 bonds, depending on whether they will carry oxygen with them or not (see Supplementary Fig. 13).

The efficiency of radiolysis in rutile $TiO_2$ can be determined from analyzing crack-filling image series at each of the different beam energies. Using the cross-section for radiolysis[2]:

$$\sigma_r(E_0) = 8\pi a_0^2 \times \left( Z \frac{U_R}{m_0 c^2} \right) \left( \frac{U_R}{E_{th}\beta^2} \right) \times \zeta \tag{1}$$

where $a_0$ is the Bohr radius, $Z$ is the atomic number of a moving unit, $U_R$ is Rydberg energy, $m_0$ is the rest mass of electron, $c$ is the speed of light, $E_{th}$ is the threshold energy that must be transferred to produce a movement, and $\beta = \frac{v}{c} = \sqrt{1 - \left(1 + \frac{E_0}{m_0 c^2}\right)^{-2}}$ and fitting to the data, the efficiency of a single step rolling in the "2-step" octahedral rolling model is estimated to be $\zeta \simeq 6 \times 10^{-6}$ (see Fig. 4e). This efficiency value is slightly lower than that determined for silicates ($\zeta \sim 10^{-4}$)[2], which is expected given that the octahedral unit in rutile $TiO_2$ is larger than the tetrahedral unit in silica. Since the cross-section for radiolysis is inversely proportional to the incident beam energy, the speed of the octahedral motion can be accelerated by order of magnitude using a lower-energy electron beam (~10 keV). Based on information about crystal structure, excitonic energy and lifetime, band gap, and atomic bond dissociation energies (summarized in Supplementary Table 2), other oxides – anatase $TiO_2$, rutile $SnO_2$ and $GeO_2$, even α-quartz $SiO_2$ and $Al_2O_3$ – have all the essential ingredients to exhibit similar radiolysis-driven restructuring via rolling building blocks. However, carefully designed experiments are needed to confirm these predictions.

In conclusion, analysis of high-resolution HAADF-STEM images combined with core-level EELS spectra shows that high-energy (80-300 keV) electron beams at high doses ($\gtrsim 10^7$ e/nm²) can radiolytically restructure a crystalline material instead of amorphizing it. Such unusual constructive radiolysis-driven restructuring was observed in rutile $TiO_2$. The sample geometry, wherein nanometer-scale cracks with atomically sharp edges are aligned with the STEM beam, allowed atomic-resolution visualization of this radiolysis-driven restructuring of the surrounding rutile $TiO_2$ crystal. We propose that this restructuring is the result of many "2-step rolling" movements of the octahedral units located at the exposed corners, edges, and surfaces of the crack. Channeling radiolysis as a constructive force sheds new light on atomistic mechanisms that drive radiolytic structural modifications in insulating materials. It should be noted that based on observations discussed above and estimations of the cross-sections for knock-on sputtering of the O and Ti atoms (see Supplementary Fig. 14), at the beam energies above 350 keV, sputtering of atoms should be considerable and at energies above 650 keV be dominant. Therefore, electron beams with energies above 350 keV should be avoided. Since the average time needed to flip "octahedrons" is in seconds for these doses rates and electron energies (see Supplementary Fig. 15), such radiolysis-based crystal restructuring can be effectively tuned by varying beam energy and dose rate to fit a wide range of the growth techniques and growth rates. We predict that similar radiolysis-driven restructuring should occur in other oxides with crystal structures consisting of octahedral or tetrahedral building blocks, as long as they satisfy the requirements for significant radiolysis. These observations point to new possibilities for using an electron beam to treat sharp cracks in brittle ceramics, improve the quality of wide bandgap thin films during the growth, and engineer novel nanostructures with atomic precision.

## Methods

### IrO₂/TiO₂ growth and nano cracks formation

Samples were prepared by growing a 26 nm $IrO_2$ film on a rutile $TiO_2$ (001) substrate using solid-source metal-organic molecular beam epitaxy (SSMOMBE)[36]. The rutile $TiO_2$ (001) substrate was annealed in oxygen plasma for 20 mins at growth temperature before $IrO_2$ film growth. Ir was supplied by sublimation of 99.9 % pure Ir (acac)₃ (American Elements), an air stable solid metal-organic precursor, which was placed in a crucible (E-Science, Inc.) inside a custom-built low temperature effusion cell (E-Science, Inc.). The effusion cell temperature was set to be 175 °C for Ir supply. A radio frequency plasma source (Mantis) with charge deflection plates, operated at a forward power of 250 W was used for supplying reactive oxygen species required for Ir oxidation. To avoid formation of oxygen vacancies and surface decomposition of $IrO_2$ to Ir metal, all films were cooled to 120 °C after growth in the presence of oxygen plasma. During the $IrO_2$ film growth on the $TiO_2$ substrate, atomically sharp cracks formed due to epitaxial strain ($\varepsilon = 2.2\%$ in $\langle 110 \rangle$ direction)[24]. These cracks span micrometers along the in-plane $\langle 110 \rangle$ crystallographic directions and propagate well into the $TiO_2$ substrate along the $c$-axis.

### SEM and STEM characterization

SEM images were acquired with a 2 kV and 25 pA current electron beam using JEOL 6500 FEG-SEM. For STEM characterization, electron-transparent cross-sectional lamella were prepared using Focused Ion Beam (FIB) using FEI Helios NanoLab G4 dual-beam system. A ~50 nm a-C layer was pre-deposited on the film using a carbon sputter coater to protect the whole film from electron beam and ion beam exposure. Additional layers of a-C (2 μm) and Pt (2 μm) were deposited on the region of interest before ion beam trenching. Ga ion beam FIB was operated at 30 kV with ion beam current ranging from 7 pA to 9.1 nA.

STEM experiments were performed on an aberration-corrected FEI Titan G2 60–300 (S)TEM microscope equipped with a CEOS DCOR probe corrector, a Schottky field emission gun, a monochromator, a super-X energy dispersive X-ray (EDX) spectrometer, and a Gatan Enfinium ER spectrometer. HAADF-STEM images were acquired at 200 kV with 50 pA probe current. The STEM beam screen current, calibrated with Faraday cup, was set to be 50 pA without sample (through the vacuum). Then dose rate was determined as screen current divided by the area exposed. For image acquisition, a dwell time of 16 μs per pixel was chosen and the total time for each frame was determined accordingly. Camera length was set to be 130 mm with probe convergence angle of 25.5 mrad for HAADF-STEM imaging. The detector inner and outer collection angles were 55 and 200 mrad respectively. STEM-EELS data of Ti $L_{2,3}$- and O K- edges was obtained using a monochromated probe with energy resolution of 0.13–0.14 eV at the energy dispersion of 0.1 eV. STEM-EELS probe convergence angle was 25.5 mrad and STEM camera length was set to be 38 mm for EELS acquisition. Dual EELS mode was used to measure both low-loss region with zero-loss peak (ZLP) and the core-loss region. For thickness determination of the specific location of the rutile-$TiO_2$ sample, the "log-ratio" method was applied to the low-loss EELS spectrum[25]. The mean free path of plasmon excitation in rutile $TiO_2$ is $\lambda_p = 130$ nm[37].

### STEM images simulation

These images are simulated using *Multislice* method and TEMSIM code developed by Kirkland[38] with following STEM parameters: $E_O = 200$ keV, $C_{S(3)} = 0$, $\Delta f = 0$, $\alpha_{obj} = 25$ mrad, and HAADF detector inner and outer angles of 50 and 200 mrad. These values are used to mimic the condition of the STEM used for these experiments. Thermal atomic displacement values of 0.075 Å for Ti and 0.110 Å for O atoms are used for $T = 300$ K[39,40]. Ten frozen phonon configurations are averaged for each image. The final simulated images were convoluted with 2D Gaussian function with the full width at half maximum (FWHM) of 1 Å to incorporate the effects of source size[41].

## Data availability

All the essential data required to evaluate the conclusions in this paper has been provided in the article and the Supplementary Information. To ensure transparency and facilitate further research, Source Data file has been deposited in DRUM database with free access under accession code: https://doi.org/10.13020/ya9y-bp34[42].

## Code availability

The codes used to perform calculations and fittings are performed using MATLAB® and are available from the authors upon request.

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

## Acknowledgements

This work was supported primarily by the National Science Foundation (NSF) through the University of Minnesota MRSEC under Award No. DMR –2011401 (B.J. and K.A.M.) and partially through NSF Grant No. DMR –2309431 (K.A.M.). Parts of this work were carried out at the UMN Characterization Facility, supported in part by the NSF through the UMN MRSEC program. The authors acknowledge the Minnesota Supercomputing Institute (MSI) at the University of Minnesota for providing computational resources. Film growth was supported by the U.S. Department of Energy (DOE) through Grant No. DE-SC0020211 (B.J.). We wish to thank Dr. Michael Odlyzko for help with STEM experiments and providing feedback.

## Author contributions

S.G. and K.A.M. conceived the project. S.G. performed STEM experiments and analyzed data with input from H.Y. and K.A.M. S.N. grew thin films under the guidance of B.J. S.G. and K.A.M. prepared the manuscript with contributions from all authors. K.A.M. directed all aspects of the project.

## Competing interests

The authors declare no competing interests.
