## [Peer Review File · Nature Communications]

Mending Cracks Atom-by-atom in Rutile TiO₂ with Electron Beam RadiolysisREVIEWER COMMENTS

Reviewer #1 (Remarks to the Author):

This work reported an interesting discovery that direct visualization of radiolysis-driven restructuring instead of amorphizing of rutile TiO₂ under electron beam irradiation. The progressive filling of atomically sharp nanometer-wide cracks of rutile TiO₂ was revealed by STEM. And a “two-step rolling” model of mobile octahedral building blocks was introduced to explain the radiolysis-driven atomic migration. The data reported here can support the conclusions. I think this work provide a route for the engineering of novel nanostructures atom-by-atom by using the electron probes and therefore is recommend for publication after minor revisions.

1. What will happen for the rutile TiO₂ under higher electron dose? Amorphizing? Or reduced?
2. How does the Ti³⁺ turn into Ti⁴⁺ after reforming new TiO₂ crystal structure?
3. Will Ti-O bonds be broken and Ti metal ions be reduced for the location without cracks of rutile TiO₂ under electron beam irradiation? If yes, will the Ti atom be driven to construct new TiO₂ crystal structure or to make it amorphous?
4. Similar experiments for other metal oxides are suggested to be provided to prove the general nature of this new strategy if possible.

Reviewer #2 (Remarks to the Author):

In this paper, Guo et al. directly observed radiolysis-driven restructuring of the rutile TiO₂ crystal having nanometer-scale cracks with atomically sharp edges by HAADF-STEM imaging. First of all, the material itself which has single crystal rutile TiO₂ with IrO₂ film on it is very interesting since it has inherent atomically sharp crack along the <110> direction, which enables this interesting study. In that material, the authors carried out advanced analysis and derived radiolysis model with literature support. So I recommend publication of this paper in Nature Communication after corrections on the following specific comments.

- 1) To present the evolution of the cracks, the authors provide number of electron dose. The measurement method is recommended to be added. Did you measure the dose with Faraday method or how can you evaluate the accuracy?
- 2) The damage by electron happens through knock-on as well as radiolysis. If the income energy is high enough so the displacement by knock-on is major mechanism in rutile TiO₂, the cation will be removed instead of TiO₆ octahedra rolling behavior. In the experiment of 300 keV, the radiolysis seems to be more dominant than knock-on damage. If the authors suggest the threshold e-beam energy when the radiolysis behavior is limited rather knock-on displacement occurs, that would provide the guideline for the application of this study.
- 3) It is interesting to investigate the contribution factor of surface and bulk to the atomic bonding structure at the crack edge, explaining “self-healing”. What about the bonding structure evolution in the bridge region? I guess it would have more surface dominant structure at the early stage and then move the bulk dominant structure. Can you show the result with data? If you can provide the structure evolution at the crack edge and the brige both, it will strengthen you model.
- 4) “2-step” model assume that the TiO₆ octahedra rotate 90° and the authors categorized the interstitial Ti atoms in Fig. 4c. For this, I think it is necessary to precondition that a 90° rotation is preferred because it is energetically lower. Please provide a supporting rationale

for your use of the 90 ° degree rotation model and justify it. Also please show a few cases demonstrating how you measure the interstitial positions for “bright” and “dim” rows respectively in Fig. 4c. You explained how you define and measure the Ti interstitial atoms in Fig. S7, but they are ideal simulated cases. Please show additional figure in Fig. 4 demonstrating which Ti atoms is interstitial, why it is belong to “dim” or ”bright” rows, how to measure the distance for each cases would help the understanding of the readers.

5) Other minor things

- Needs English correction

- In Fig.3 “g” and “E” are missed. The caption does not match with the figure.

Reviewer #3 (Remarks to the Author):

See attached report

The manuscript by S. Guo *et al.* reports on the visualization of radiolysis-driven restructuring of nanoscale cracks in rutile TiO₂. An interesting model for the atomic reconstruction is offered based on the experimental STEM and EELS measurements. Descriptions of structural changes in TiO₂ driven by the electron beam has been presented by others (for instance ref. 21, 22), but the authors have performed a through analysis and examined rigorously different experimental conditions such as beam dose, accelerating voltage, etc. to study this local reconstruction. The atomic reconstruction from crystal to crystal within nanoscale distances is fascinating and it represents a fundamental knowledge which can further deepen our understanding of oxide crystal growth/engineering and the illusive radiolysis mechanisms. There is no doubt that this work highlights an important piece of atomic restructuring, and it has important novelty elements. I enjoyed reading this manuscript very much. The manuscript is very well organized, well written and contains fair amount of information (supplementary information) to be understood by the audience.

There is however a consideration about structural integrity for the “2-step rolling” model which has puzzled me a bit. I was wondering whether the integrity of the TiO₆ octahedron block unit is fully preserved during rotation. Can other configurations assisted by the electron beam and surface diffusion become more stable? The authors should comment on this apparent preservation of structural integrity and how other potential configurations are ruled out, if any.

Theoretical calculations (e.g. molecular dynamics) might shed some light into potential intermediate configurations between the initial and final stages of rotation of the octahedron block, and contribute to validate the suggested rolling model. The authors might want to offer a comment on the challenges to simulate dynamical process of nanostructures driven by radiolysis and future investigation directions.

It would also be worthy to comment on the experimental limitations to study the temporal evolution of the octahedral unit “rotation” and the temporal scale of those suggested rotations.

The comments and questions above do not change my positive impression about the submitted manuscript. I recommend publication of this manuscript after the authors have addressed the comments/questions. Finally, the authors might want to reconsider changing the article title since “atom-by-atom” does not necessarily emphasize the structural integrity of the octahedron unit rotation.

REVIEWER COMMENTS

Reviewer #1 (Remarks to the Author):

This work reported an interesting discovery that direct visualization of radiolysis-driven restructuring instead of amorphizing of rutile TiO₂ under electron beam irradiation. The progressive filling of atomically sharp nanometer-wide cracks of rutile TiO₂ was revealed by STEM. And a “two-step rolling” model of mobile octahedral building blocks was introduced to explain the radiolysis-driven atomic migration. The data reported here can support the conclusions. I think this work provide a route for the engineering of novel nanostructures atom-by-atom by using the electron probes and therefore is recommend for publication after minor revisions.

1. What will happen for the rutile TiO₂ under higher electron dose? Amorphizing? Or reduced?

Response: These are good questions. We perform additional experiments in bulk regions of rutile TiO₂ as well as in crack regions with continuous e-beam exposure to very high doses reaching $1.4 \times 10^7 \text{ e}/\text{\AA}^2$. While the crack region keeps healing, as expected, the bulk region starts to show signs of surface restructuring or maybe amorphization. Fig. R1 below shows three panels of the same bulk region at three exposure doses. As can be seen below, the sample starts to lose its original rutile structure at very high doses. It is noticeable in both the images as well as the corresponding fast Fourier transforms (FFTs). It could be due to amorphization, or restructuring, or both.

Fig. R1

Changes made: With the reviewer’s comment in mind, we added this Fig. R1 into the Supporting Information (Fig. S9) with caption. We also added one sentence about this in the main text and refer to this figure.

2. How does the Ti^{3+} turn into Ti^{4+} after reforming new TiO_2 crystal structure?

Response: The rolling “octahedrons” that are located on the surfaces, edges and corners have primarily Ti^{3+} as these units are combination of mostly TiO_5 units (due to required Ti-O bond breakage as discussed in the text) mixed with some TiO_4 and TiO_6 units. When such two “octahedrons” meet at the center of the crack after migrating from each side, they lock with each other and form rutile TiO_2 structure. The oxygen atoms at meeting sites are now shared by locked “octahedrons” and the units become true TiO_6 octahedrons. As a result, the TiO_5 (with Ti^{3+}) “octahedrons” become TiO_6 octahedrons (with Ti^{4+}). A simplified schematic below (Fig. R2) illustrates how these TiO_x units (numbered 1 to 4) with some missing oxygen (the red dashed open circles) rotate and lock with other “octahedrons” from the other side. As they now share oxygens, each unit becomes complete $Ti^{4+}O_6$ octahedron.

Fig. R2

Changes made: We added this simplified schematic into Supporting Information (Fig. S4b) with caption to describe the final step of “octahedron” locking and transition of Ti cations from Ti^{3+} to Ti^{4+} . We also refer to this figure in the main text.

3. Will Ti-O bonds be broken and Ti metal ions be reduced for the location without cracks of rutile TiO_2 under electron beam irradiation? If yes, will the Ti atom be driven to construct new TiO_2 crystal structure or to make it amorphous?

Response: For the locations without cracks (or away from the cracks), while bond breakage would still be happening, we haven’t observed any materials loss or structural modifications for same doses. We measured it using quantitative EELS. As can be seen in the Fig. R3 below, for the same irradiation doses as the crack region, the amount of O and Ti atoms in areas away from crack are practically unchanged.

Fig. R3

With such high binding energy between Ti and O atoms (3.3 eV per bond, Table S2), only single-bonded (dangling) O atoms on the surface of rutile TiO₂ have a chance to be sputtered away (Fig. R4 below for sputtering cross-sections). All other atoms (double- or triple-bonded oxygen or 4-, 5-, or 6-bonded Ti) will not be sputtered. Also, as discussed in the main text and shown in SI Fig. S13, surface migration of “octahedral” units is extremely unlikely, as it requires many (3 to 8, Fig. S13c) bond-breakages. However, since TEM sample preparation unavoidably creates some surface roughness with edges and corners, we expect and have observed some light surface restructuring in those areas away from the crack regions under extremely high doses as shown in Fig. R1.

Fig. R4

Changes made: With the reviewer’s comment in mind, we have now included the bulk integrated intensity and the sputtering cross-sections shown above into Supporting Information (Fig. S8 and Fig. S14, correspondingly) with captions. We also refer to these two figures in the main text.

4. Similar experiments for other metal oxides are suggested to be provided to prove the general nature of this new strategy if possible.

Response: While the suggestion is good and worth consideration, doing similar experiments in other oxides or in other systems is not possible at this moment. As was discussed in the main text, the key for this systematic study is availability of such TiO₂ samples with nano-cracks (e.g., atomically sharp surfaces with nanometer gaps). We don't have access to other samples of such nature, nor are we even aware of where to obtain those. However, we agree with reviewer here that such study will be worthwhile in the future.

Reviewer #2 (Remarks to the Author):

In this paper, Guo et al. directly observed radiolysis-driven restructuring of the rutile TiO₂ crystal having nanometer-scale cracks with atomically sharp edges by HAADF-STEM imaging. First of all, the material itself which has single crystal rutile TiO₂ with IrO₂ film on it is very interesting since it has inherent atomically sharp crack along the <110> direction, which enables this interesting study. In that material, the authors carried out advanced analysis and derived radiolysis model with literature support. So I recommend publication of this paper in Nature Communication after corrections on the following specific comments.

1) To present the evolution of the cracks, the authors provide number of electron dose. The measurement method is recommended to be added. Did you measure the dose with Faraday method or how can you evaluate the accuracy?

Response: The electron doses in these experiments were measured using fluorescent screen of our FEI Titan STEM, and the fluorescent screen current was calibrated with Faraday cup. The STEM beam screen current was set to be 50 pA without sample (through the vacuum). Then dose rate was determined as screen current divided by the area exposed. For image acquisition, a dwell time of 16 μ s per pixel was chosen and the total time for each frame was determined accordingly.

Changes made: With the reviewer's comment in mind, we added these experimental details into the Methods section.

2) The damage by electron happens through knock-on as well as radiolysis. If the incoming energy is high enough so the displacement by knock-on is major mechanism in rutile TiO₂, the cation will be removed instead of TiO₆ octahedra rolling behavior. In the experiment of 300 keV, the radiolysis seems to be more dominant than knock-on damage. If the authors suggest the threshold e-beam energy when the radiolysis behavior is limited rather knock-on displacement occurs, that would provide the guideline for the application of this study.

Response: This is a very good comment. Our experimental observations (shown in Fig. 3d in the main text, Fig. R3 above) as well as calculated surface sputtering cross-sections shown in Fig. R4, show that the radiolysis dominates up to 300 keV of e-beam energy. Based on the intersections of the calculated cross-sections for sputtering (Fig. S14 in SI) and the fitted cross-section for radiolysis (Fig. 4e in main text), we predict that starting around 350 keV the knock-on damage will be considerable, and will be dominant above 650 keV in rutile TiO₂ and, therefore, should be avoided.

Changes made: With the reviewer's comment in mind, we added the above Fig. R3 and Fig. R4 discussed above with captions into Supporting Information Fig. S8 and Fig. S14. We also included statement in the conclusion sections in the main text with guideline for application referring to this figure.

3) It is interesting to investigate the contribution factor of surface and bulk to the atomic bonding structure at the crack edge, explaining “self-healing”. What about the bonding structure evolution in the bridge region? I guess it would have more surface dominant structure at the early stage and then move the bulk dominant structure. Can you show the result with data? If you can provide the structure evolution at the crack edge and the bridge both, it will strengthen you model.

Response: These are indeed very insightful questions. Yes, we do have atomic-resolution HAADF-STEM images exactly capturing what reviewer is asking here. Figure below, which is now in Supporting Information, clearly shows modifications in the atomic structure of the rutile TiO_2 at the edges of crack during healing. Yellow dotted line highlights the regions with higher-level of modifications. Magnified images from these regions, including bridging and non-bridging sites (labeled A to C), show close-up of the structural modifications. In particular, panel ‘C’ shows how “bright rows” become less bright with added intensities between original rutile columns. They are indicated by yellow arrows.

Fig. R5

Changes made: We added brief discussion into the main text and referred to this SI figure (Fig. S11a). We added more clarifications into the caption of this figure. To further improve the visualizations of the structure of healing crack, we included a 3D model below (Fig. R6) to SI (Fig. S4a).

Fig. R6

4) “2-step” model assume that the TiO_6 octahedra rotate 90° and the authors categorized the interstitial Ti atoms in Fig. 4c. For this, I think it is necessary to precondition that a 90° rotation is preferred because it is energetically lower. Please provide a supporting rationale for your use of the 90° degree rotation model and justify it. Also please show a few cases demonstrating how you measure the interstitial positions for “bright” and “dim” rows respectively in Fig. 4c. You explained how you define and measure the Ti interstitial atoms in Fig. S7, but they are ideal simulated cases. Please show additional figure in Fig. 4 demonstrating which Ti atoms is interstitial, why it is belong to “dim” or “bright” rows, how to measure the distance for each cases would help the understanding of the readers.

Response: This comment has two parts. We address them separately.

Part 1. At every 90° rotation, the octahedrons at the corners, edges and surfaces can lock with neighboring units without distortion (stretching, squiring, bending, etc.), as those positions of octahedrons are crystallographic allowed and available. They will share corner oxygen atoms and lower the surface energy. However, these locations are interstitial crystallographic locations in rutile TiO_2 . This rational and crystallography of this 90° rotations of octahedra units can also be visualized from top view (view along rotation axis), which is presented in Fig. S10. Any other rotation $< 90^\circ$ results in dangling octahedra unit. and any rotation $> 90^\circ$ is prohibited by crystal structure of rutile TiO_2 .

Part 2. We provided an example of the method used to determine the positions interstitial Ti (or octahedra unit) and its distance from the reference column in SI Fig. S11b (a case of “Bright row”). The statistical analysis of these positions is presented in Fig. 4d in the main text. Having the reviewer’s comment here, we added more examples (Fig. R7 shown below) in SI Fig. S11c highlighting cases for both, “bright row” and “dim row”.

Fig. R7

Changes made: We have expanded SI Fig. S11 and added examples shown above. The caption to Fig. S11 is expanded accordingly.

5) Other minor things

- Needs English correction
- In Fig.3 “g” and “E” are missed. The caption does not match with the figure.

Response: We would like to thank the reviewer for catching these typos and errors.

Changes made: We have fixed them.

Reviewer #3 (Remarks to the Author):

The manuscript by S. Guo et al. reports on the visualization of radiolysis-driven restructuring of nanoscale cracks in rutile TiO₂. An interesting model for the atomic reconstruction is offered based on the experimental STEM and EELS measurements. Descriptions of structural changes in TiO₂ driven by the electron beam has been presented by others (for instance ref. 21, 22), but the authors have performed a thorough analysis and examined rigorously different experimental conditions such as beam dose, accelerating voltage, etc. to study this local reconstruction. The atomic reconstruction from crystal to crystal within nanoscale distances is fascinating and it represents a fundamental knowledge which can further deepen our understanding of oxide crystal growth/engineering and the illusive radiolysis mechanisms. There is no doubt that this work highlights an important piece of atomic restructuring, and it has important novelty elements. I enjoyed reading this manuscript very much. The manuscript is very well organized, well written and contains fair amount of information (supplementary information) to be understood by the audience.

1) There is however a consideration about structural integrity for the “2-step rolling” model which has puzzled me a bit. I was wondering whether the integrity of the TiO₆ octahedron block unit is fully preserved during rotation. Can other configurations assisted by the electron beam and surface diffusion become more stable? The authors should comment on this apparent preservation of structural integrity and how other potential configurations are ruled out, if any.

Response: The reviewer’s observation is correct here. The rolling “octahedron” is not always a complete octahedron. It cannot be. The process of breaking bonds and rolling units results in many cases leaving behind shared corner oxygen atoms. Based on number of bonds that needs to be broken (shown in Fig. S13c) and the probability of shared oxygen “go” or “stay”, we concluded that the majority of rolling “octahedrons” are TiO₅, along with some TiO₆ and TiO₄ units. That is exactly why in the main text we have the sentence: “*When a Ti-O bond is broken in rutile TiO₂ and the O atom is left behind, the original TiO₆ reduces to a TiO₅ “octahedron” in the and the Ti⁴⁺ turns into Ti³⁺.*”, and, in the discussion of rolling units, we used terms “TiO₅ “octahedra”” or “TiO_x “octahedra””. Octahedron in prentices is used to indicate that they are not always true TiO₆ octahedrons. Also, in the Figs. 4a,b and S10 we used open circle instead of solid red sphere to indicate that the oxygen there is missing. We suspect that not indicating this in the captions clearly may have contributed to this confusion.

Changes made: Having the reviewer’s comment in mind, we modified the main text to clarify this point. We also expanded captions to Fig. 4 in the main text and Fig. S10 in SI to clarify that the open circles indicate missing oxygen atoms.

2) Theoretical calculations (e.g. molecular dynamics) might shed some light into potential intermediate configurations between the initial and final stages of rotation of the octahedron block, and contribute to validate the suggested rolling model. The authors might want to offer a comment on the challenges to simulate dynamical process of nanostructures driven by radiolysis and future investigation directions.

Response: We concur with reviewer here. Theoretical calculation would indeed shed more light into these observations. After several in-depth discussions with colleagues, we came to conclusion that such theoretical study is quite resource and time intensive. Proper theoretical study will require both DFT-based calculation for structures and energetics, and atomistic simulations for rolling dynamics. While we are indeed interested in these calculations and will pursue them in the near future (as soon as recourses are available), at this stage it is not doable for us in timely manner.

3) It would also be worthy to comment on the experimental limitations to study the temporal evolution of the octahedral unit “rotation” and the temporal scale of those suggested rotations.

Response: This is an insightful comment. Since we have experimental data for different beam energies and doses, we have evaluated the temporal scale for these radiolytic “octahedron” unit rotations. The figure below summarizes the results. Using these results, we estimated the experimental temporal limits for different voltages and doses. The results are discussed in the revised main text.

Fig. R8

Changes made: With the reviewer’s comment here in mind, we added one more figure into Supporting Information (Fig. S15) with corresponding caption. We added a short discussion in the main text on this topic with reference to this SI Figure.

4) Finally, the authors might want to reconsider changing the article title since “atom-by-atom” does not necessarily emphasize the structural integrity of the octahedron unit rotation.

Response: We would like to thank review for the suggestion. We thought about this. Considering the fact that structural integrity of the “octahedrons” is not preserved at all time during rolling, having “atom-by-atom” in the title would be more suitable.

REVIEWERS' COMMENTS

Reviewer #1 (Remarks to the Author):

I am satisfied with the changes made by authors.

Reviewer #2 (Remarks to the Author):

I generally agree and understand the answers sent by the authors. And the authors presented quite a lot of supplementary data to the reviewers' questions, and the answers to my comments were appropriately reflected in the main manuscript. So I will accept this paper as it is.

Reviewer #3 (Remarks to the Author):

Authors addressed properly my questions and comments. Authors used referee's feedback to improve article clarity and result richness. I recommend this manuscript for publication.

REVIEWER COMMENTS and OUR RESPONCES

Reviewer #1:

Comment:

I am satisfied with the changes made by authors.

Response:

We appreciate the reviewer for positive comment.

Reviewer #2:

Comment:

I generally agree and understand the answers sent by the authors. And the authors presented quite a lot of supplementary data to the reviewers' questions, and the answers to my comments were appropriately reflected in the main manuscript. So I will accept this paper as it is.

Response:

We appreciate the reviewer for positive comment.

Reviewer #3:

Comment:

Authors addressed properly my questions and comments. Authors used referee's feedback to improve article clarity and result richness. I recommend this manuscript for publication.

Response:

We appreciate the reviewer for positive comment.